# Continual Named Entity Recognition without Catastrophic Forgetting

**Duzhen Zhang**[1,2*], **Wei Cong**[1,2*], **Jiahua Dong**[1,2†], **Yahan Yu**[3], **Xiuyi Chen**[3],
**Yonggang Zhang**[4] and **Zhen Fang**[5]

[1]Shenyang Institute of Automation, Chinese Academy of Sciences, Shenyang, China
[2]University of Chinese Academy of Sciences, Beijing, China
[3]Baidu Inc, Beijing, China
[4]Department of Computer Science, Hong Kong Baptist University, HongKong, China
[5]Australian Artificial Intelligence Institute, University of Technology Sydney, Sydney, Australia
{bladedancer957,congwei45,dongjiahua1995}@gmail.com

## Abstract

Continual Named Entity Recognition (CNER) is a burgeoning area, which involves updating an existing model by incorporating new entity types sequentially. Nevertheless, continual learning approaches are often severely afflicted by catastrophic forgetting. This issue is intensified in CNER due to the consolidation of old entity types from previous steps into the non-entity type at each step, leading to what is known as the semantic shift problem of the non-entity type. In this paper, we introduce a pooled feature distillation loss that skillfully navigates the trade-off between retaining knowledge of old entity types and acquiring new ones, thereby more effectively mitigating the problem of catastrophic forgetting. Additionally, we develop a confidence-based pseudo-labeling for the non-entity type, *i.e.,* predicting entity types using the old model to handle the semantic shift of the non-entity type. Following the pseudo-labeling process, we suggest an adaptive re-weighting type-balanced learning strategy to handle the issue of biased type distribution. We carried out comprehensive experiments on ten CNER settings using three different datasets. The results illustrate that our method significantly outperforms prior state-of-the-art approaches, registering an average improvement of 6.3% and 8.0% in Micro and Macro F1 scores, respectively.[1]

## 1 Introduction

Named Entity Recognition (NER) is a essential research area in Natural Language Understanding (NLU). Its purpose is to assign each token in a sequence with multiple entity types or non-entity type (Ma and Hovy, 2016). Recently, the advent of Pre-training Language Models (PLMs) (Kenton and Toutanova, 2019) has ushered NER into a

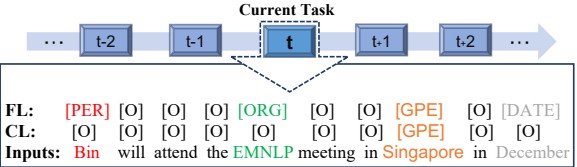

Figure 1: A simplified CNER example, where **FL** and **CL** denote Full ground-truth Labels and Current ground-truth Labels, respectively. Old entity types (such as [ORG] (Organization), [PER] (Person)) and future entity types (such as [DATE] (Date)) are masked as [O] (the non-entity type) at the current step $t$ where [GPE] (Countries) is the current entity type to be learned, causing the semantic shift problem of the non-entity type (the second row **CL**).

new epoch. Conventionally, NER operates within a paradigm where tokens are classified into some fixed entity types (such as Organization, Person, etc.), and the NER model undergoes a one-time learning process. However, a more realistic scenario calls for the NER model to continually identify novel entity types as they emerge, without the need for a complete retraining. This paradigm, known as Continual Named Entity Recognition (CNER), has gained substantial research attention due to its promising practical applications (Monaikul et al., 2021; Zheng et al., 2022). A case in point is voice assistants like Siri and Xiao Ai, which are frequently required to extract new entity types (such as Genre, Actor) to understand new user intents (for example, GetMovie).

Deep learning approaches to CNER encounter two primary challenges. The first one is common to all continual learning methods, known as catastrophic forgetting (Robins, 1995; McCloskey and Cohen, 1989; Goodfellow et al., 2013; Kirkpatrick et al., 2017). This refers to the tendency of neural networks to forget previously acquired knowledge when learning new information. Existing CNER methods (Monaikul et al., 2021; Xia et al., 2022; Zheng et al., 2022) often resort to the widely em-

---

*Equal contributions.
†The corresponding author is Dr. Jiahua Dong.
[1]Our code is available at https://github.com/BladeDancer957/CPFD.

ployed technique of knowledge distillation (Hinton et al., 2015) to tackle catastrophic forgetting. However, these methods meticulously extract the output probabilities for previous entity types from the old model and transfer them to the new model. Consequently, they tend to imbue the model with excessive stability (*i.e.*, retention of old information) at the expense of plasticity (*i.e.*, acquisition of new information).

The second one is specific to CNER, involving the semantic shift of the non-entity type. In the conventional NER paradigm, tokens are marked as the non-entity type, indicating that they do not belong to any entity type. In contrast, in the CNER paradigm, tokens marked as the non-entity type imply that they do not belong to any of the current entity types. This implies that the non-entity type may encompass: the true non-entity type, previously learned old entity types, or future ones not yet encountered. As depicted in Figure 1 (the second row **CL**), the old entity types `Organization` and `Person` (learned in the prior steps such as $t$-1, $t$-2, etc.) as well as the future entity type `Date` (will be learned in the future steps such as $t$+1, $t$+2, etc.) are all marked as the non-entity type at the current step $t$. Lacking mechanisms to differentiate tokens that pertain to previous entity types from the authentic non-entity type, this semantic shift could worsen catastrophic forgetting. To our knowledge, Zheng et al. were the first to recognize the old entity types contained within the non-entity type using the old model to distill causal effect. However, they don't sufficiently handle recognition errors by the old model.

In this paper, we present a novel approach named CPFD, an acronym for Confidence-based pseudo-labeling and Pooled Features Distillation, which utilizes the old model in two significant ways to address the aforementioned challenges inherent in CNER. Firstly, we introduce a pooled features distillation loss that strikes a judicious trade-off between stability and plasticity, thus effectively alleviating catastrophic forgetting. These features are grounded in the attention weights learned by PLMs, capturing crucial linguistic knowledge necessary for the NER task, including coreference and syntax information (Clark et al., 2019). Secondly, we develop a confidence-based pseudo-labeling strategy to specifically identify previous entity types within the current non-entity type for classification, mitigating the problem of semantic shift. To better re-

duce the recognition errors from the old model, we employ entropy as a measure of uncertainty and the median entropy as a confidence threshold, retaining only those pseudo labels where the old model exhibits sufficient confidence. Furthermore, we introduce an adaptive re-weighting type-balanced learning strategy to handle the biased distribution between new and old entity types that occurs post pseudo-labeling. This approach adaptively assigns different weights to the tokens of different types based on the number of tokens. Our contributions can be summarized as follows:

- We design a pooled features distillation loss to alleviate catastrophic forgetting by retaining linguistic knowledge and establishing a suitable balance between stability and plasticity.

- We develop a confidence-based pseudo-labeling strategy to better recognize previous entity types for the current non-entity type tokens and deal with the semantic shift problem. To cope with the imbalanced type distribution, we propose an adaptive re-weighting type-balanced learning strategy for CNER.

- Extensive results on ten CNER settings of three datasets indicate that our CPFD achieves remarkable improvements over the existing State-Of-The-Art (SOTA) approaches with an average gain of 6.3% and 8.0% in Micro and Macro F1 scores, respectively.

## 2 Related Work

**Continual Learning** learns continuous tasks without reducing performance on previous tasks (Chen and Liu, 2018; Dong et al., 2022, 2023b). We divide existing continual learning methods into memory-based, dynamic architecture-based, and regularization-based. Memory-based methods (Lopez-Paz and Ranzato, 2017; Rebuffi et al., 2017; Shin et al., 2017) learn a new task by integrating the saved or generated old samples into the current training samples. Dynamic architecture-based methods (Mallya and Lazebnik, 2018; Rosenfeld and Tsotsos, 2018; Yoon et al., 2018) dynamically extend the model architecture to learn new tasks. Regularization-based methods impose constraints on network weights (Aljundi et al., 2018; Kirkpatrick et al., 2017; Zenke et al., 2017), intermediary features (Hou et al., 2019a), or output probabilities (Li and Hoiem, 2017; Dong et al., 2023a) for relieving catastrophic forgetting.

**CNER** Traditional NER focuses on the development of various deep learning models aimed at extracting entities from unstructured text (Li et al., 2020). Recently, PLMs have been widely utilized in NER and have achieved SOTA performance (Kenton and Toutanova, 2019; Liu et al., 2019). However, most existing methods are designed to recognize a fixed set of predefined entity types. In response, CNER incorporates the continual learning paradigm with traditional NER (Zhang et al., 2023b; Ma et al., 2023; Zhang et al., 2023a). ExtendNER (Monaikul et al., 2021) explores the application of knowledge distillation to CNER, wherein the new model learns to fit the classifier logits of the old model. L&R (Xia et al., 2022) offers a 'learn-and-review' framework for CNER. The learning stage mirrors ExtendNER, while the reviewing stage generates synthetic samples of previous entity types to enhance the current dataset. CFNER (Zheng et al., 2022) introduces a causal framework for CNER and distills causal effects from the non-entity type. In spite of significant advancements, these methods strictly distill the output probabilities for previous types from the old model into the new model, resulting in excessive stability but limited plasticity in the models. Moreover, these approaches do not adequately address the semantic shift problem. CFNER employs the old model to recognize non-entity type tokens belonging to previous entity types for distilling causal effect and utilizes a curriculum learning strategy to mitigate recognition error. However, this curriculum learning strategy requires the manual pre-definition of numerous hyper-parameters, limiting its applicability and the effect of error reduction.

In contrast, we propose a pooled features distillation loss. This retains crucial linguistic knowledge embedded within attention weights and strikes a suitable balance between the stability and plasticity of the model, thus more efficiently mitigating catastrophic forgetting issue. Moreover, we design a confidence-based pseudo-labeling strategy for classification. This strategy employs median entropy as the confidence threshold, eliminating the need for manually predefining any hyper-parameters, better reducing the recognition error from the old model, and dealing with the semantic shift problem.

## 3 Preliminary

CNER aims to train a model across $t = 1, ..., T$ steps, progressively learning an expanding set of entity types. Each step has its unique training set $\mathcal{D}_t$, comprising multiple pairs $(X^t, \boldsymbol{Y}^t)$, where $X^t$ represents an input token sequence with a length of $|X^t|$ and $\boldsymbol{Y}^t$ represents the corresponding ground truth label sequence encoded in a one-hot format. Notably, $\boldsymbol{Y}^t$ only includes labels for the current entity types $\mathcal{E}^t$, with all other labels (for example, future entity types $\mathcal{E}^{t+1:T}$ or potential old entity types $\mathcal{E}^{1:t-1}$) collapsed into the non-entity type $e_o$. At step $t$ ($t>1$), considering the old model $\mathcal{M}_{t-1}$ as well as the current training set $\mathcal{D}_t$, our objective is to update a novel model $\mathcal{M}_t$ capable of recognizing entities from all types seen thus far, represented by $\bigcup_{i=1}^{t} \mathcal{E}^i$.

## 4 Method

In the above formulation, we pinpoint two significant challenges in CNER. The first one is the issue of catastrophic forgetting (French, 1999), which implies that the neural network may entirely and abruptly forget previous entity types $\mathcal{E}^{1:t-1}$ when learning the current types $\mathcal{E}^t$. This issue is further amplified by the second challenge: semantic shift. At step $t$, tokens labeled as the non-entity type are intrinsically ambiguous, as they could include the true non-entity type, previously learned old entity types, or as-yet-unseen future entity types. To address these challenges in CNER, we propose a novel CPFD method, which takes advantage of the previous model in two ways, shown in Figure 2.

### 4.1 Pooled Features Distillation

Recent studies (Clark et al., 2019) suggest that attention weights learned by PLMs can encapsulate rich linguistic knowledge, including coreference and syntactical information, both critical to NER. In light of this, we first propose a feature distillation loss designed to encourage the transfer of linguistic knowledge contained within attention weights from the previous model to the new model, formulated as follows:

$$\mathcal{L}_{\text{FD}} = \sum_{k=1}^{K} \sum_{i=1}^{|X^t|} \sum_{j=1}^{|X^t|} \left|\left| \boldsymbol{A}_{\ell,k,i,j}^t - \boldsymbol{A}_{\ell,k,i,j}^{t-1} \right|\right|^2 \quad (1)$$

where $\boldsymbol{A}_\ell^t$ and $\boldsymbol{A}_\ell^{t-1} \in \mathbb{R}^{K \times |X^t| \times |X^t|}$ correspond to the attention weights of layer $\ell$ for $\mathcal{M}_{t-1}$ and $\mathcal{M}_t$ respectively, $\ell = 1, ..., L$, with $K$ standing for the count of attention heads.

However, both the output probabilities distillation found in prior CNER methods (Monaikul et al.,

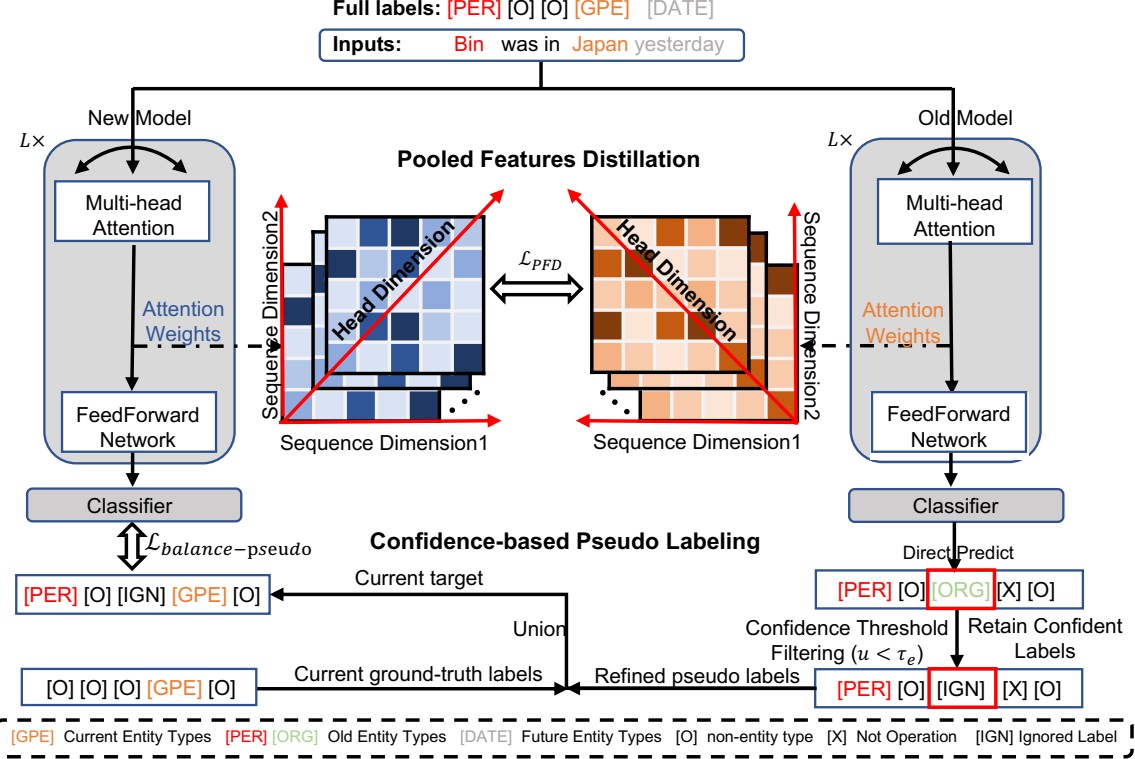

Figure 2: Our CPFD method aims to learn a NER model within a continual learning paradigm, where old entity types are collapsed into the non-entity type in the current step. We constitute a suitable balance between stability and plasticity by pooled features distillation loss to prevent catastrophic forgetting and generate high-quality pseudo-labels from old predictions by a confidence-based pseudo-labeling strategy to deal with the semantic shift problem.

2021; Xia et al., 2022; Zheng et al., 2022) and our feature distillation delineated in Equation (1) can be considered rigid losses. This approach compels $\mathcal{M}_t$ to strictly align with the output probabilities or attention weights found in $\mathcal{M}_{t-1}$. This practice often results in an overabundance of stability (*i.e.*, preservation of prior knowledge), consequently impairing its plasticity (*i.e.*, ability to assimilate new knowledge). Contemporary studies further endorse the notion that the distillation operation ought to establish a balance between stability and plasticity (Douillard et al., 2020, 2021; Pelosin et al., 2022; Kurmi et al., 2021).

To this goal, we incorporate a pooling operation into our proposed loss, thus permitting a level of plasticity by consolidating the pooled dimensions (Douillard et al., 2020; Pelosin et al., 2022). The reasoning for this approach is that the minimal possible plasticity imposes an absolute similarity between the old and new model, whereas increased plasticity relaxes this definition of similarity. Pooling can lessen this similarity, and more aggressive pooling can offer a greater degree of freedom.

By pooling the sequence dimensions, we derive

a more permissive loss that preserves only the head dimension, formulated as follows:

$$\mathcal{L}_{\text{PFD-lax}} = \sum_{k=1}^{K} \left\| \sum_{i,j=1}^{|X^t|} \boldsymbol{A}_{\ell,k,i,j}^{t} - \sum_{i,j=1}^{|X^t|} \boldsymbol{A}_{\ell,k,i,j}^{t-1} \right\|^2 \tag{2}$$

Our proposed pooling-based framework facilitates the formulation of a more flexible feature distillation loss, striking an improved balance between plasticity and stability. Based on Equation (2), we can moderately sacrifice plasticity to enhance stability through less aggressive pooling, achieved by aggregating statistics across only one of the head and sequence dimensions:

$$\begin{aligned} \mathcal{L}_{\text{PFD}} = &\sum_{i=1}^{|X^t|} \sum_{j=1}^{|X^t|} \left\| \sum_{k=1}^{K} \boldsymbol{A}_{\ell,k,i,j}^{t} - \sum_{k=1}^{K} \boldsymbol{A}_{\ell,k,i,j}^{t-1} \right\|^2 \\ + &\sum_{k=1}^{K} \sum_{j=1}^{|X^t|} \left\| \sum_{i=1}^{|X^t|} \boldsymbol{A}_{\ell,k,i,j}^{t} - \sum_{i=1}^{|X^t|} \boldsymbol{A}_{\ell,k,i,j}^{t-1} \right\|^2 \\ + &\sum_{k=1}^{K} \sum_{i=1}^{|X^t|} \left\| \sum_{j=1}^{|X^t|} \boldsymbol{A}_{\ell,k,i,j}^{t} - \sum_{j=1}^{|X^t|} \boldsymbol{A}_{\ell,k,i,j}^{t-1} \right\|^2 \end{aligned} \tag{3}$$

$\mathcal{L}_{\text{PFD}}$ achieves an appropriate balance between excessive rigidity (as demonstrated by Equation (1)) and extreme leniency (as shown in Equation (2)).

## 4.2 Confidence-based Pseudo-labeling

As previously noted, tokens marked as the non-entity type at step $t$ might actually belong to the authentic non-entity type, previous entity types, or future entity types. Simply categorizing these tokens as the non-entity type could intensify catastrophic forgetting. To tackle this issue of semantic shift linked to the non-entity type, we design a pseudo-labeling strategy (Lee et al., 2013; Douillard et al., 2021). Specifically, we utilize the predictions of old model for non-entity type tokens as indications of their true entity types, especially whether they are related to any of the previously acquired types.

Formally, we represent the cardinality of the current entity types with $E^t = \text{card}(\mathcal{E}^t)$. We denote the current model's predictions, encompassing the true non-entity type, all the old entity types, and the current ones, with $\widehat{\boldsymbol{Y}}^t \in \mathbb{R}^{|X^t| \times (1+E^1+...+E^t)}$. We define $\widetilde{\boldsymbol{Y}}^t \in \mathbb{R}^{|X^t| \times (1+E^1+...+E^t)}$ as the target at step $t$, calculated using the one-hot ground-truth label sequence $\boldsymbol{Y}^t \in \mathbb{R}^{|X^t| \times (1+E^1+...+E^t)}$ in step $t$ and pseudo-labels extracted from the predictions of the old model $\widehat{\boldsymbol{Y}}^{t-1} \in \mathbb{R}^{|X^t| \times (1+E^1+...+E^{t-1})}$. The process is described as follows:

$$\widetilde{Y}_{i,e}^t = \begin{cases} 1 \text{ if } \boldsymbol{Y}_{i,e_o}^t = 0 \ \& \ e = \underset{e' \in \mathcal{E}^t}{\text{argmax}} \ \boldsymbol{Y}_{i,e'}^t \\ 1 \text{ if } \boldsymbol{Y}_{i,e_o}^t = 1 \ \& \ e = \underset{e' \in e_o \cup \mathcal{E}^{1:t-1}}{\text{argmax}} \ \widehat{Y}_{i,e'}^{t-1} \\ 0 \text{ otherwise} \end{cases} \quad (4)$$

In other words, if a token is not marked as the non-entity type $e_o$, we replicate the ground truth label. Otherwise, we utilize the label predicted by the old model. This pseudo-labeling strategy enables the assignment of the actual semantic label to each token labeled as the non-entity type, provided the token belongs to any of the previous entity types. Nevertheless, labeling all non-entity type tokens as pseudo-labels can be unproductive, *e.g.,* on uncertain tokens where the old model is likely to falter. Thus, this Vanilla Pseudo-Labeling (VPL) strategy inadvertently propagates errors from the previous model's incorrect predictions to the new model.

Inspired by (Saporta et al., 2020), we propose a Confidence-based Pseudo-Labeling (CPL) strategy, designed to diminish the influence of label noises and effectively tackle the semantic shift problem. This strategy employs entropy as a measure of uncertainty, with the median entropy serving as the

confidence threshold. Specifically, before embarking on learning step $t$, we initially input $\mathcal{D}_t$ into $\mathcal{M}_{t-1}$ for inference and calculate the entropy $u$ of the output distribution related to each non-entity type token. Subsequently, for each non-entity type token, we can categorize these tokens according to their predicted old type (which corresponds to the index of the largest output probability predicted by the previous model). This allows us to calculate the median entropy $\tau_e$ for old type $e \in e_o \cup \mathcal{E}^{1:t-1}$ within each group. Following this, we can modify Equation (4) as follows:

$$\widetilde{Y}_{i,e}^t = \begin{cases} 1 \text{ if } \boldsymbol{Y}_{i,e_o}^t = 0 \ \& \ e = \underset{e' \in \mathcal{E}^t}{\text{argmax}} \ \boldsymbol{Y}_{i,e'}^t \\ 1 \text{ if } \boldsymbol{Y}_{i,e_o}^t = 1 \ \& \ e = \underset{e' \in e_o \cup \mathcal{E}^{1:t-1}}{\text{argmax}} \ \widehat{Y}_{i,e'}^{t-1} \ \& \ u < \tau_e \\ 0 \text{ otherwise} \end{cases}$$
$$(5)$$

where $u$ denotes the uncertainty of token $i$ as well as $\tau_e$ is a type-specific confidence threshold. Equation (5) only retains pseudo-labels where the previous model is "*confident*" enough ($u < \tau_e$).

Following the pseudo-labeling process, we find that the count of new-type tokens present in current sequences generally exceeds the count of pseudo-labeled old-type tokens. This type-imbalance issue typically skews the updated classifier towards new types. Inspired by (Zhao et al., 2022), we introduce an Adaptive Re-weighting Type-balanced (ART) learning strategy that adaptively assigns varying weights to tokens of different types based on their quantity. Thus, the balanced pseudo-labeling cross-entropy loss can be expressed as follows:

$$\mathcal{L}_{\text{balance-pseudo}} = -\frac{1}{|X^t|} \sum_i^{|X^t|} \eta_i \widetilde{\boldsymbol{Y}}_i^t \log \widehat{\boldsymbol{Y}}_i^t \quad (6)$$

where $\eta_i$ denotes the weight of the token at the location $i$ in the sequence $X^t$, computed as follow:

$$\eta_i = \begin{cases} 0.5 + \sigma(\dfrac{N^{\text{old}}}{N^{\text{new}}}) & \text{if } \widetilde{\boldsymbol{Y}}_i^t \in \mathcal{E}^{1:t-1} \\ 1.0 & \text{otherwise} \end{cases} \quad (7)$$

where $N^{\text{old}}$, $N^{\text{new}}$ and $\sigma(\cdot)$ are the number of tokens belonging to old entity types $\mathcal{E}^{1:t-1}$, the number of tokens belonging to the new entity types $\mathcal{E}^t$ and the sigmoid function, respectively.

Finally, the total loss in CPFD is:

$$\mathcal{L}(\Theta_t) = \underbrace{\mathcal{L}_{\text{balance-pseudo}}}_{\text{classification}} + \lambda \underbrace{\mathcal{L}_{\text{PFD}}}_{\text{distillation}} \quad (8)$$

with $\lambda$ a hyper-parameter for balancing losses, and $\Theta_t$ is the set of learnable parameters for $\mathcal{M}_t$.

Table 1: The statistics for each dataset.

| | # Entity Type | # Sample | Entity Type Sequence (Alphabetical Order) |
|---|---|---|---|
| CoNLL2003 | 4 | 21k | LOCATION, MISC, ORGANISATION, PERSON |
| I2B2 | 16 | 141k | AGE, CITY, COUNTRY, DATE, DOCTOR, HOSPITAL, IDNUM, MEDICALRECORD, ORGANIZATION, PATIENT, PHONE, PROFESSION, STATE, STREET, USERNAME, ZIP |
| OntoNotes5 | 18 | 77k | CARDINAL, DATE, EVENT, FAC, GPE, LANGUAGE, LAW, LOC, MONEY, NORP, ORDINAL, ORG, PERCENT, PERSON, PRODUCT, QUANTITY, TIME, WORK_OF_ART |

## 5 Experiments

### 5.1 Experimental Setup

To ensure fair comparisons with SOTA approaches, we follow the experimental setup outlined in CFNER (Zheng et al., 2022).

**Datasets** We conduct the evaluation of CPFD using three widely adopted NER datasets: CoNLL2003 (Sang and De Meulder, 1837), I2B2 (Murphy et al., 2010), and OntoNotes5 (Hovy et al., 2006). The statistics of these three datasets are specifically presented in Table 1.

We divide the training set into disjoint slices, each corresponding to different continual learning steps, employing the same sampling algorithm as suggested in CFNER (Zheng et al., 2022). Within each slice, we exclusively keep the labels belonging to the entity types under learn, while others are masked as the non-entity type. For more comprehensive explanations and the detailed breakdown of this sampling algorithm, refer to Appendix B of CFNER (Zheng et al., 2022).

**CNER Settings** In terms of training, we introduce entity types in the same alphabetical order as CFNER (Zheng et al., 2022) and learn models sequentially with corresponding data slices. In particular, FG entity types are employed to train the base model, and for each continual learning step, we utilize PG entity types, represented as FG-a-PG-b. For the CoNLL2003 dataset, we apply two CNER settings: FG-1-PG-1 and FG-2-PG-1. For the I2B2 and OntoNotes5 datasets, we establish four CNER settings: FG-1-PG-1, FG-2-PG-2, FG-8-PG-1, and FG-8-PG-2. During evaluation, we maintain only the labels of the current entity types to learn, masking others as the non-entity type within the validation set. At each step, the model yielding the best validation performance is chosen for both testing and the next step. In the testing phase, we preserve labels for all previously learned entity types, designating the rest as the non-entity type within the test set.

**Performance Metrics** Consistent with CFNER, we utilize Micro F1 (Mi-F1) and Macro F1 score (Ma-F1) to evaluate the model's performance, taking into account the issue of entity type imbalance in NER. We present the mean result across all steps, encompassing the first, as the ultimate performance. Further, to offer a more detailed analysis, we introduce step-wise performance comparison line plots. To assess the statistical significance of the improvements, we perform a paired t-test with a significance level of 0.05.

**Baseline Methods** We benchmark our CPFD against the recent CNER methods, namely Extend-NER (Monaikul et al., 2021) and CFNER (Zheng et al., 2022), with the latter serving as the previous SOTA method. We also draw comparisons with continual learning methodologies employed in the field of computer vision, such as Self-Training (ST)(Rosenberg et al., 2005; Lange et al., 2019), LUCIR(Hou et al., 2019b), and POD-Net (Douillard et al., 2020). Zheng et al. successfully adapted these three methods to CNER settings. Additionally, we evaluate fine-tuning (FT) without the integration of any anti-forgetting practices, thereby establishing a lower bound for comparison. For a more comprehensive introduction to these baselines, please consult Appendix C of CFNER (Zheng et al., 2022).

**Implementation Details** In alignment with prior CNER methods (Monaikul et al., 2021; Xia et al., 2022; Zheng et al., 2022), we adopt the "BIO" labeling schema for all datasets. Our NER model utilizes the bert-base-cased (Kenton and Toutanova, 2019) model as the encoder, featuring a layer depth ($L$) of 12 and an attention head count ($K$) of 12, and employs a fully-connected layer as the classifier. We implement the model using the PyTorch framework (Paszke et al., 2019), built atop the BERT Huggingface implementation (Wolf et al., 2019). For each setting, if PG=1, we train the model for 10 epochs; otherwise, for 20 epochs. We establish the batch size, learning rate, and balancing weight $\lambda$ as 8, $4e$-4, and 2, respectively. In the total loss, we also penalize the KL divergence between the new model's classifier logits against the old model's classifier logits to overcome forgetting better (Monaikul et al., 2021). All experiments are carried out on an NVIDIA A100 GPU with 40GB of memory, and each experiment is run 5 times to ensure statistical robustness.

Table 2: Comparisons with baselines on I2B2 and OntoNotes5. The **red** denotes the highest result, and the **blue** denotes the second highest result. The marker † refers to significant test $p\text{-}value < 0.05$ comparing with CFNER. $*$ represents results from re-implementation. Other baseline results are cited from CFNER (Zheng et al., 2022).

| Dataset | Baseline | FG-1-PG-1 | | FG-2-PG-2 | | FG-8-PG-1 | | FG-8-PG-2 | |
| --- | --- | --- | --- | --- | --- | --- | --- | --- | --- |
| | | Mi-F1 | Ma-F1 | Mi-F1 | Ma-F1 | Mi-F1 | Ma-F1 | Mi-F1 | Ma-F1 |
| I2B2 | FT | 17.43±0.54 | 13.81±1.14 | 28.57±0.26 | 21.43±0.41 | 20.83±1.78 | 18.11±1.66 | 23.60±0.15 | 23.54±0.38 |
| | PODNet | 12.31±0.35 | 17.14±1.03 | 34.67±2.65 | 24.62±1.76 | 39.26±1.38 | 27.23±0.93 | 36.22±12.9 | 26.08±7.42 |
| | LUCIR | 43.86±2.43 | 31.31±1.62 | 64.32±0.76 | 43.53±0.59 | 57.86±0.87 | 33.04±0.39 | 68.54±0.27 | 46.94±0.63 |
| | ST | 31.98±2.12 | 14.76±1.31 | 55.44±4.78 | 33.38±3.13 | 49.51±1.35 | 23.77±1.01 | 48.94±6.78 | 29.00±3.04 |
| | ExtendNER* | 41.65±10.11 | 23.11±2.70 | 67.60±1.15 | 42.58±1.59 | 45.14±2.91 | 27.41±0.88 | 56.48±2.41 | 38.88±1.38 |
| | ExtendNER | 42.85±2.86 | 24.05±1.35 | 57.01±4.14 | 35.29±3.38 | 43.95±2.01 | 23.12±1.79 | 52.25±5.36 | 30.93±2.77 |
| | CFNER* | 64.79±0.26 | 37.79±0.65 | 72.58±0.59 | 51.71±0.84 | 56.66±3.22 | 36.84±1.35 | 69.12±0.94 | 51.61±0.87 |
| | CFNER | 62.73±3.62 | 36.26±2.24 | 71.98±0.50 | 49.09±1.38 | 59.79±1.70 | 37.30±1.15 | 69.07±0.89 | 51.09±1.05 |
| | **CPFD (Ours)** | 74.19±0.95† | 48.34±1.45† | 78.19±0.58† | 56.04±1.22† | 74.75±1.35† | 56.19±2.46† | 81.05±0.87† | 65.04±1.13† |
| | **Imp.** | ⇑9.40 | ⇑10.55 | ⇑5.61 | ⇑4.33 | ⇑14.96 | ⇑18.89 | ⇑11.93 | ⇑13.43 |
| OntoNotes5 | FT | 15.27±0.26 | 10.85±1.11 | 25.85±0.11 | 20.55±0.24 | 17.63±0.57 | 12.23±1.08 | 29.81±0.12 | 20.05±0.16 |
| | PODNet | 9.06±0.56 | 8.36±0.57 | 19.04±1.08 | 16.93±0.85 | 29.00±0.86 | 20.54±0.91 | 37.38±0.26 | 25.85±0.29 |
| | LUCIR | 28.18±1.15 | 21.11±0.84 | 56.40±1.79 | 40.58±1.11 | 66.46±0.46 | 46.29±0.38 | 76.17±0.09 | 55.58±0.55 |
| | ST | 50.71±0.79 | 33.24±1.06 | 68.93±1.67 | 50.63±1.66 | 73.59±0.66 | 49.41±0.77 | 77.07±0.62 | 53.32±0.63 |
| | ExtendNER* | 51.36±0.77 | 33.38±0.98 | 63.03±9.39 | 47.64±5.15 | 73.65±0.19 | 50.55±0.56 | 77.86±0.10 | 55.21±0.51 |
| | ExtendNER | 50.53±0.86 | 32.84±0.84 | 67.61±1.53 | 49.26±1.49 | 73.12±0.93 | 49.55±0.90 | 76.85±0.77 | 54.37±0.57 |
| | CFNER* | 58.44±0.71 | 41.75±1.51 | 72.10±0.31 | 55.02±0.35 | 78.25±0.33 | 58.64±0.42 | 80.09±0.37 | 61.06±0.37 |
| | CFNER | 58.94±0.57 | 42.22±1.10 | 72.59±0.48 | 55.96±0.69 | 78.92±0.58 | 57.51±1.32 | 80.68±0.25 | 60.52±0.84 |
| | **CPFD (Ours)** | 66.73±0.70† | 54.12±0.30† | 74.33±0.30† | 57.75±0.35† | 81.87±0.47† | 65.52±1.05† | 83.38±0.18† | 66.27±0.75† |
| | **Imp.** | ⇑7.79 | ⇑11.90 | ⇑1.74 | ⇑1.79 | ⇑2.95 | ⇑6.88 | ⇑2.70 | ⇑5.21 |

Figure 3: Comparison of the step-wise Mi-F1 on I2B2 and OntoNotes5. The result of baselines is directly cited from CFNER (Zheng et al., 2022).

## 5.2 Results and Analysis

**Comparisons with Baselines**   To substantiate the efficacy of our CPFD method across various CNER settings, we conduct exhaustive experiments on the CoNLL2003, I2B2, and OntoNotes5 datasets. The results obtained from the I2B2 and OntoNotes5 datasets are presented in Table 2, and a step-wise comparison of Mi-F1 scores is depicted in Figure 3. Due to the limitations in space, the step-wise Ma-F1 comparison results for I2B2 and OntoNotes5 datasets are provided in Figure 8 within our Appendix A. As for the outcomes obtained from the smaller dataset, CoNLL2003, these are detailed in Table 4, Figure 6 (step-wise Mi-F1 comparisons), and Figure 7 (step-wise Ma-F1 comparisons), all available in our Appendix A.

As indicated in Tables 2 and 4, our CPFD method significantly surpasses the previous SOTA method, CFNER, yielding enhancements ranging from 1.33% to 14.96% in Mi-F1 and from 0.83% to 18.89% in Ma-F1 across ten CNER settings of three datasets. Figures 3, 6, 7, and 8 further corroborate that our CPFD outshines other CNER baseline methods in almost all step-wise comparisons under the ten settings. These results validate CPFD's superior performance in learning a robust CNER model, demonstrating enhanced resilience against catastrophic forgetting and semantic shift problems. Additionally, we have visually represented some qualitative comparison results from the FG-8-PG-2 setting on the OntoNotes5 dataset in Figure 4. These results further validate the efficacy of our CPFD in learning new entity types consecutively.

**Input Sentence** In 1985 , the Chinese Mentally Impaired and Blind Sports Associations were also established .
**ExtendNER PL** [O] [B-DATE] [O] [B-ORG] [I-WOA] [I-WOA] [I-WOA] [I-WOA] [I-WOA] [I-ORG] [O] [O] [O] [O]
**CFNER PL** [O] [B-DATE] [O] [B-ORG] [O] [O] [O] [O] [I-ORG] [I-ORG] [I-ORG] [O] [O] [O] [O]
**CPFD PL (Ours)** [O] [B-DATE] [O] [B-ORG] [B-NORP] [I-ORG] [I-ORG] [I-ORG] [I-ORG] [I-ORG] [I-ORG] [O] [O] [O] [O]
**Golden Labels** [O] [B-DATE] [O] [B-ORG] [I-ORG] [I-ORG] [I-ORG] [I-ORG] [I-ORG] [I-ORG] [I-ORG] [O] [O] [O] [O]

**Input Sentence** Holding the 6th Asia - Pacific Special Olympic Games in Beijing will motivate society to better support disabled sports .
**ExtendNER PL** [O] [B-WOA] [B-ORD] [I-WOA] [I-WOA] [I-WOA] [I-WOA] [I-WOA] [I-WOA] [O] [B-GPE] [O] [O] [O] [O] [O] [O] [O] [O]
**CFNER PL** [O] [O] [B-ORD] [I-EVE] [I-WOA] [I-WOA] [I-EVE] [I-WOA] [I-EVE] [O] [B-GPE] [O] [O] [O] [O] [O] [O] [O] [O]
**CPFD PL (Ours)** [O] [O] [B-ORD] [I-EVE] [I-EVE] [I-EVE] [I-EVE] [I-EVE] [I-EVE] [O] [B-GPE] [O] [O] [O] [O] [O] [O] [O] [O]
**Golden Labels** [O] [O] [B-ORD] [B-EVE] [I-EVE] [I-EVE] [I-EVE] [I-EVE] [I-EVE] [O] [B-GPE] [O] [O] [O] [O] [O] [O] [O] [O]

Figure 4: Two real cases sampled from the OntoNotes5 test set. **PL** denotes the predicted labels. **B-** and **I-** distinguish begin/inside of entities. [O], [DATE], [ORG], [WOA], [NORP], [ORD], [EVE], and [GPE] denote the non-entity type, Date, Organization, Work of art, Nationalities, Religious, or Political groups, Ordinals, Event, and Countries, Cities, or States, respectively. All the prediction results correspond to the final step of FG-8-PG-2. These visualized cases highlight the superiority and effectiveness of our CPFD method.

**Input Sentence** The Australia side will provide China with a technical cooperation grant of 20 million Australian dollars .
**CL** [O] [O] [O] [O] [O] [O] [O] [O] [O] [O] [O] [B-MON] [I-MON] [I-MON] [I-MON] [O]
**CL + VPL** [O] [B-GPE] [O] [O] [I-GPE] [B-GPE] [B-DATE] [O] [O] [I-LAN] [O] [O] [B-MON] [I-MON] [I-MON] [I-MON] [O]
**CL + CPL** [O] [B-GPE] [O] [O] [IGN] [B-GPE] [B-DATE] [O] [O] [IGN] [O] [O] [B-MON] [I-MON] [I-MON] [I-MON] [O]

Figure 5: Visualization of pseudo labels on OntoNotes5 under FG-8-PG-2. **CL** denotes current ground-truth labels. **VPL** and **CPL** denote pseudo labels generated by vanilla and confidence-based pseudo-labeling strategies. **B-** and **I-** distinguish begin/inside of entities. [IGN] denotes ignored labels. [GPE] (Countries, Cities, or States), [LAN] (Language), and [DATE] (Date) denote the old entity types in the first step. [MON] (Money) denotes the new entity type being learned in the second step. The visualization result shows that our **CPL** can reduce label noises and combine with **CL** to form a better target for the current model, compared with **VPL**.

Table 3: The ablation study of our CPFD on I2B2 and OntoNotes5 under the setting FG-1-PG-1. When compared with Ours, all ablation variants severely degrade CNER performance. It verifies the importance of all components to address CNER collaboratively.

| Methods | I2B2 | | OntoNotes5 | |
|---|---|---|---|---|
| | Mi-F1 | Ma-F1 | Mi-F1 | Ma-F1 |
| **CPFD (Ours)** | **74.19±0.95** | **48.34±1.45** | **66.73±0.70** | **54.12±0.30** |
| w/ $\mathcal{L}_{FD}$ | 71.46±1.19 | 45.17±1.28 | 63.80±1.01 | 51.83±0.73 |
| w/ $\mathcal{L}_{PFD\text{-}lax}$ | 70.22±0.90 | 43.89±1.10 | 62.32±0.53 | 50.12±0.70 |
| w/o $\mathcal{L}_{PFD}$ | 68.66±0.88 | 42.28±0.79 | 60.80±0.86 | 48.94±1.38 |
| w/o CPL | 54.86±5.36 | 37.39±3.58 | 59.37±0.82 | 46.68±0.45 |
| w/o ART | 72.29±1.56 | 45.35±1.83 | 65.19±1.33 | 52.94±0.46 |

**Ablation Study** This subsection examines the effectiveness of individual components in our CPFD method through ablation studies, the results of which are shown in Table 3. We evaluate several alternatives for the PFD loss, defined in Section 4.1. Broadly, we find that substituting our $\mathcal{L}_{PFD}$ loss with a more strict $\mathcal{L}_{FD}$ loss or a more permissive $\mathcal{L}_{PFD\text{-}lax}$ loss in our CPFD method tends to diminish CNER performance. This occurs because these two variations either omit pooling or implement a more aggressive pooling, resulting in excessive sta-

bility or plasticity in the CNER model. Our $\mathcal{L}_{PFD}$ loss presents a judicious balance between stability and plasticity, which helps to better mitigate catastrophic forgetting, without which poor CNER performance will result. **CPFD w/o CPL** denotes the absence of pseudo-labeling strategy. These results suggest that pseudo labels help the model recall what it has learned from the non-entity type tokens. Our CPL strategy can effectively reduce prediction errors from the old model and address semantic shift by retaining only confident pseudo labels. A visualization of these pseudo labels is shown in Figure 5. The results of **CPFD w/o ART** underline that ART contributes positively to addressing issues with biased type distribution.

## 6 Conclusion

In this paper, we lay the foundation for future research in CNER, an emerging field in NLU. We pinpoint two principal challenges in CNER: catastrophic forgetting and the semantic shift problem of the non-entity type. To address these issues, we first introduce a pooled feature distillation loss that carefully establishes the balance between stability and plasticity, thereby better alleviating catas-

trophic forgetting. Subsequently, we present a confidence-based pseudo-labeling strategy to explicitly extract old entity types contained in the current non-entity type, better reducing the impact of label noise and dealing with the semantic shift problem. We evaluate CPFD on ten CNER settings across three datasets and demonstrate that CPFD significantly outperforms the previous SOTA methods across all settings.

## Limitations

Our pooled features distillation loss necessitates additional computational effort to align with the intermediary features of the old model. Our confidence-based pseudo-labeling strategy, which employs median entropy as the confidence threshold, necessitates pre-calculation for each old entity type based on the current training set and the old model, thus extending the training duration. Moreover, although our confidence-based pseudo-labeling strategy helps reduce the prediction errors of the old model, it is not entirely foolproof, and some mislabeled instances may still persist.

## Ethics Statement

In relation to ethical considerations, we provide the following clarifications: (1) Our research does not engage with any sensitive data or tasks. (2) All experiments are conducted on existing datasets that have been sourced from publicly available scientific research. (3) We offer comprehensive descriptions of the dataset statistics and the hyper-parameter configurations for our method. Our analyses align with the experimental results. (4) In the interest of promoting reproducibility, we plan to make our code accessible via GitHub.

## Acknowledgement

We express our gratitude to the anonymous reviewers for their valuable and insightful comments. This research received partial support from the National Nature Science Foundation of China under Grants 32070000 and 62133005.

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

# A    Supplementary Experimental Results

Table 4: Comparisons with baselines on CoNLL2003. The red denotes the highest result, and the blue denotes the second highest result. The markers † and ‡ refer to significant tests comparing with CFNER and LUCIR, respectively ($p\text{-}value < 0.05$). ∗ represents results from re-implementation. Other baseline results are directly cited from CFNER (Zheng et al., 2022).

| Baseline | FG-1-PG-1 | | FG-2-PG-1 | |
|---|---|---|---|---|
| | Mi-F1 | Ma-F1 | Mi-F1 | Ma-F1 |
| FT | 50.84±0.10 | 40.64±0.16 | 57.45±0.05 | 43.58±0.18 |
| PODNet | 36.74±0.52 | 29.43±0.28 | 59.12±0.54 | 58.39±0.99 |
| LUCIR | 74.15±0.43 | 70.48±0.66 | 80.53±0.31 | 77.33±0.31 |
| ST | 76.17±0.91 | 72.88±1.12 | 76.65±0.24 | 66.72±0.11 |
| ExtendNER* | 76.07±0.35 | 73.06±0.29 | 77.89±0.42 | 69.92±1.02 |
| ExtendNER | 76.36±0.98 | 73.04±1.80 | 76.66±0.66 | 66.36±0.64 |
| CFNER* | 80.29±0.21 | 78.44±0.24 | 81.52±0.43 | 77.20±0.82 |
| CFNER | 80.91±0.29 | 79.11±0.50 | 80.83±0.36 | 75.20±0.32 |
| **CPFD (Ours)** | **82.24±0.63†** | **79.94±0.66†** | **85.70±0.19†** | **83.49±0.16‡** |
| **Imp.** | ⇑**1.33** | ⇑**0.83** | ⇑**4.18** | ⇑**6.16** |

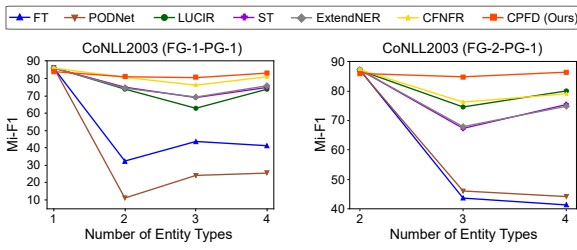

Figure 6: Comparison of the step-wise Mi-F1 on CoNLL2003. The result of baselines is directly cited from CFNER (Zheng et al., 2022).

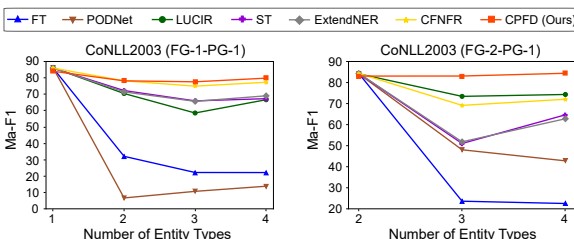

Figure 7: Comparison of the step-wise Ma-F1 on CoNLL2003. The result of baselines is directly cited from CFNER (Zheng et al., 2022).

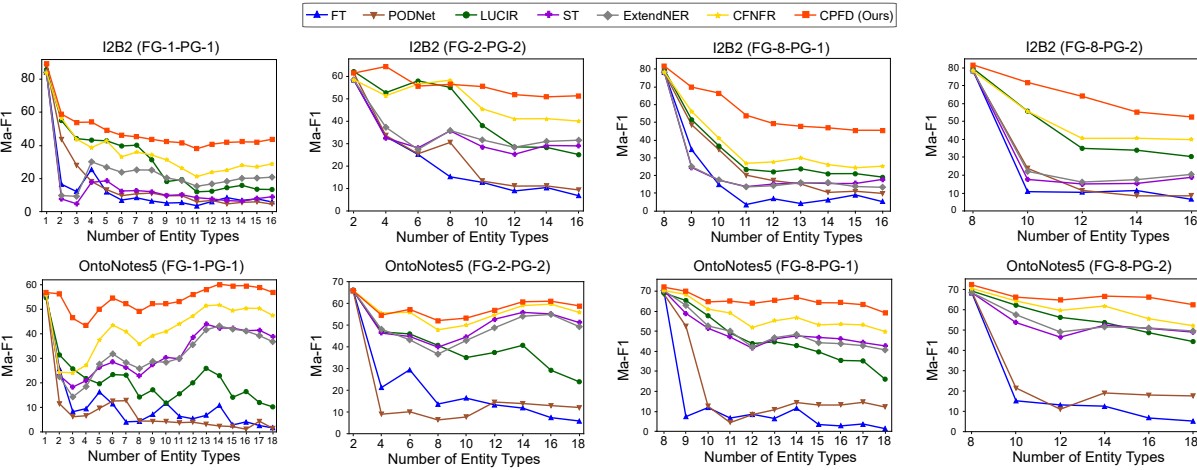

Figure 8: Comparison of the step-wise Ma-F1 on I2B2 and OntoNotes5. The result of baselines is directly cited from CFNER (Zheng et al., 2022).