# OpenReview forum: "Continual Named Entity Recognition without Catastrophic Forgetting"
_EMNLP/2023/Conference — EMNLP 2023 Main_

### Official Review · Reviewer_xCyC · 2023-07-19

**Typos Grammar Style And Presentation Improvements:** 1. The caption of Figure 2 is a littl…
**Soundness:** 4

**Excitement:**

4: Strong: This paper deepens the understanding of some phenomenon or lowers the barriers to an existing research direction.

**Missing References:**

None

**Paper Topic And Main Contributions:**

This paper lays the foundation for future investigations in CNER, a novel area within NLU. The primary focus of this research is to address two key challenges in CNER: catastrophic forgetting and the semantic shift problem of the non-entity type.To tackle these challenges, this paper proposes pooled feature distillation loss and confidence-based pseudo-labeling. This paper evaluates conduct experiments on ten CNER settings across three datasets and show that the proposed method significantly outperforms the previous SOTA methods consistently.

**Questions For The Authors:**

Qestion A: Why using the entropy to measure the confidence of the old model? Can we use the maximum prediction score as the metric to select pseudo-labels?
Qestion B: Why distiling the attention score matrix instead of other model parameters, e.g., the projection matrices in self attention or MLP?
Qestion C: The paper identifies the type-imbalance issue where the classifier tends to predict new classes. Why it is important to address this issue? What are the differences and connections between the type-imbalance and the class-imbalance issue.

==============================================================

Post-rebuttal:
Most of my concerns have been addressed.

**Reasons To Accept:**

1. The motivation is clear: this paper focuses on the semantic shift problem in Conintual Named Entity Recogntion, which is overlooked by previous methods.
2. The comparison are extensive: this paper compares the proposed method with other baselines on different settings and datasets.

**Reasons To Reject:**

1. This paper only conduct experiments on BERT, the performance on larger models is unclear.
2. The performance under different hyper-parameter settings (e.g., \lambda) is unclear.

**Reproducibility:**

4: Could mostly reproduce the results, but there may be some variation because of sample variance or minor variations in their interpretation of the protocol or method.

**Reviewer Confidence:**

4: Quite sure. I tried to check the important points carefully. It's unlikely, though conceivable, that I missed something that should affect my ratings.

---

> ### Author Rebuttal · Authors · 2023-08-26
>
> Thanks for your thoughtful reviews. We would like to address your concerns and answer your questions in the following.
>
> Q1: This paper only conduct experiments on BERT, the performance on larger models is unclear.
>
> A1: Thank you for your valuable feedback. We have included experiments on larger models, including bert-large-cased and roberta large. The results on the I2B2 dataset under the setting FG-8-PG-2 are as follows:
>
> | Backbone | Avg. Mi-F1 | Avg. Ma-F1|
> | -------- | -------- | -------- |
> | bert-base-cased (original paper) | 81.05 $\pm$ 0.87 | 65.04 $\pm$ 1.13 |
> | bert-large-cased | 82.04 $\pm$ 0.54 | 66.98 $\pm$ 0.83|
> | roberta-large  | **82.58 $\pm$ 0.91** | **67.61 $\pm$ 1.08** |
>
> Due to time constraints, the results of other settings will be provided in the final version. In the future, we will also attempt to use larger and more advanced pre-trained language models as the backbone encoder.
>
>
> Q2: The performance under different hyper-parameter settings (e.g., $\lambda$) is unclear.
>
> A2: Thank you for raising this question. We have conducted an analysis of hyperparameter $\lambda$ on the I2B2 dataset under the FG-8-PG-2 setting. This hyperparameter is utilized for loss balancing, and the results are as follows:
>
> | $\lambda$ | Avg. Mi-F1 | Avg. Ma-F1|
> | -------- | -------- | -------- |
> | 0.5 | 80.23 $\pm$ 0.56 | 64.59 $\pm$ 0.93 |
> | 1  |  80.12 $\pm$ 0.62 | 64.41 $\pm$ 0.78 |
> | 2  |  **81.05 $\pm$ 0.87** | **65.04 $\pm$ 1.13** |
> | 5  |  77.29 $\pm$ 0.76 | 61.82 $\pm$ 1.32 |
>
> The best performance is achieved when $\lambda$ is set to $2$.
>
>
> Q3: Why use entropy to measure the confidence of the old model? Can we use the maximum prediction score as the metric to select pseudo-labels?
>
> A3: Thank you for your thoughtful question. The use of entropy as a measure to assess the confidence of the old model's predictions stems from its ability to capture the uncertainty inherent in the prediction distribution. Low entropy indicates a state of less uncertainty, suggesting that the model is confident in its prediction. This aligns with our intention to select pseudo-labels where the old model is more certain. While using the maximum prediction score could be an alternative approach, it can be unproductive, e.g., on uncertain tokens where the old model is likely to fail. This Vanilla Pseudo-Labeling (VPL) approach more easily propagates errors from the old model's incorrect predictions to the new model. To better reduce the recognition errors from the old model, our Confidence-based Pseudo-Labeling (CPL) approach employs entropy as a measure of uncertainty and the median entropy as a confidence threshold, only retaining those pseudo labels where the old model exhibits sufficient confidence. Furthermore, the ablation experiment results in Table 2 of our paper also demonstrate the superior performance of CPL over VPL, indicating that using entropy is more effective than using the maximum prediction score.
>
>
> Q4: Why distiling the attention score matrix instead of other model parameters, e.g., the projection matrices in self attention or MLP?
>
> A4: Thank you for your insightful question. Recent studies [1] suggest that attention scores learned by pre-trained language models can encapsulate rich linguistic knowledge, including coreference and syntactical information, both critical to natural language understanding tasks like named entity recognition. In light of this, we distill the attention score matrix to encourage the transfer of linguistic knowledge from the old model to the new one. However, we appreciate your suggestion and will consider exploring the impact of distilling other model parameters (e.g., the projection matrices in self attention or MLP) in future research iterations. Your feedback is invaluable in refining our approach.
>
>
> Q5: The paper identifies the type-imbalance issue where the classifier tends to predict new classes. Why is it important to address this issue? What are the differences and connections between the type-imbalance and the class-imbalance issue.
>
> A5: Thank you for your insightful question. Following the pseudo-labeling process, we find that the count of new-type tokens present in current sequences generally exceeds the count of pseudo-labeled old-type tokens. This type-imbalance issue typically skews the updated classifier towards new types. Resolving the type-imbalance issue is crucial for creating models that are accurate, fair, and capable of making informed decisions across all entity types, regardless of their distribution. Here, the type-imbalance issue is essentially the class-imbalance issue. In alignment with prior CNER methods, we adopt the "BIO" tagging schema for all three dataset. This means that each entity type corresponds to two labels (classes): B-type and I-type, where B-/I- distinguishes the beginning/inside of an entity.
>
>
> Q6: The caption of Figure 2 is a little bit confusing. What do the "[X] Not Operation" and "[IGN] Ignored Label" mean?
>
> A6: Thank you for bringing up this point. We apologize for any confusion caused by the caption of Figure 2. The meaning of "[X] Not Operation" is that if the current token has not been labeled as the non-entity type [O] (i.e., it has been labeled as the current entity type being learned), no pseudo-labeling operation is performed. In other words, the pseudo-labeling strategy is only applied to tokens currently labeled as [O]. The meaning of "[IGN] Ignored Label" is that our confidence-based pseudo-labeling strategy retains pseudo-labels generated by the old model only if they have a sufficient level of confidence. Pseudo-labels with lower confidence will be disregarded and will not be included in the computation of the cross-entropy loss.
>
>
> **Reference**
>
> [1] Clark K, Khandelwal U, Levy O, et al. What Does BERT Look at? An Analysis of BERT’s Attention[C]//Proceedings of the 2019 ACL Workshop BlackboxNLP: Analyzing and Interpreting Neural Networks for NLP. 2019: 276-286.

---

### Official Review · Reviewer_nTza · 2023-08-03

**Soundness:** 4

**Excitement:**

3: Ambivalent: It has merits (e.g., it reports state-of-the-art results, the idea is nice), but there are key weaknesses (e.g., it describes incremental work), and it can significantly benefit from another round of revision. However, I won't object to accepting it if my co-reviewers champion it.

**Paper Topic And Main Contributions:**

This paper addresses two challenges in Chinese Named Entity Recognition (CNER): catastrophic forgetting and non-entity type semantic transfer issues. To tackle these problems, the paper adopts a phased approach and proposes corresponding strategies. It introduces an aggregated feature distillation loss to balance stability and plasticity, effectively mitigating catastrophic forgetting. Additionally, a confidence-based pseudo-labeling plan is offered to explicitly extract old entity types embedded within non-entity types, thereby reducing the impact of label noise and handling the semantic transfer problem. The paper further validates the significance of collaboration between all components in solving the CNER problem through ablation experiments.

**Reasons To Accept:**

The strengths of this paper are its novel approach to addressing the problem of catastrophic forgetting in Continual Named Entity Recognition (CNER), the extensive experiments conducted on ten CNER settings of three datasets to demonstrate the approach's effectiveness, and the significant improvements achieved over previous state-of-the-art methods. The paper's proposed approach and experimental results could inspire further research and improve performance in CNER and other related tasks.

**Reasons To Reject:**

Some potential weaknesses of this paper could include:
1. The proposed approach may not apply to all NLP tasks and datasets, and its effectiveness may depend on specific data characteristics.
2. The paper does not provide a detailed analysis of the limitations of the proposed approach, such as cases where it may not work well or situations where it may be less effective.
3. The paper does not compare the proposed approach with other state-of-the-art methods outside the CNER domain, which may limit its generalizability to other NLP tasks.
4. The paper does not provide a detailed discussion of the computational complexity of the proposed approach, which may be a concern for large-scale datasets or real-time applications.
The authors need to tell the strengths and weaknesses of their approach and provide a balanced and nuanced discussion of their findings.

**Reproducibility:**

4: Could mostly reproduce the results, but there may be some variation because of sample variance or minor variations in their interpretation of the protocol or method.

**Reviewer Confidence:**

4: Quite sure. I tried to check the important points carefully. It's unlikely, though conceivable, that I missed something that should affect my ratings.

---

> ### Author Rebuttal · Authors · 2023-08-28
>
> We appreciate your insightful feedback. We are committed to addressing your concerns and providing answers to your questions below.
>
> Q1: The proposed approach may not apply to all NLP tasks and datasets, and its effectiveness may depend on specific data characteristics.
>
> A1: Thank you for your thoughtful question. Our CPFD method consists of two parts, Confidence-based pseudo-labeling and Pooled Features Distillation. The former is designed to address specific issues in Continual Named Entity Recognition (CNER), namely the semantic shift of the non-entity type. The latter is employed to tackle the common challenge of continual learning, denoted as catastrophic forgetting. In other words, our Pooled Features Distillation (PFD) can be applied to almost all NLP continual learning tasks. In addition to the CNER task presented in our paper, we also provide results on continual text classification and continual relation learning tasks.
>
> **continual text classification**
>
> Due to time constraints, we conducted experiments on one dataset, THU-Seq, which is a task sequence constructed from THUCNews. The results are shown in the table below. All baseline results are directly quoted from [1]. Snapshot is the previous State-Of-The-Art (SOTA) method for continual text classification. $*$ represents results based on our re-implementation using open-source code.
>
> | THU-Seq:        | THU$_1$         | THU$_2$         | THU$_3$         | THU$_4$         |
> | --------------- | --------------- | --------------- | --------------- | --------------- |
> | MTL             | $99.0$          | $95.3$          | $95.9$          | $95.7$          |
> | FT              | $98.9$          | $61.2$          | $37.1$          | $27.4$          |
> | LwF             | $98.9$          | $76.8$          | $53.0$          | $43.8$          |
> | EWC             | $98.9$          | $82.5$          | $54.6$          | $51.8$          |
> | R-Walk          | $98.9$          | $84.9$          | $81.2$          | $73.5$          |
> | MAS             | $98.9$          | $86.4$          | $82.3$          | $60.1$          |
> | AdapterParallel | $98.7$          | $72.0$          | $71.6$          | $69.4$          |
> | AdapterStack    | $98.7$          | $55.6$          | $50.7$          | $33.3$          |
> | AdapterFusion   | $98.7$          | $52.7$          | $36.2$          | $26.5$          |
> | Snapshot        | $98.9{\pm0.1}$ | $87.9{\pm0.2}$ | $89.2{\pm0.4}$ | $86.3{\pm0.4}$ |
> | Snapshot + Our PFD*       | ${98.9{\pm 0.1}}$ | $\mathbf{89.3{\pm0.1}}$ | $\mathbf{90.3{\pm0.3}}$ | $\mathbf{87.6{\pm0.2}}$ |
>
>
> **continual relation learning**
>
> We conducted experiments on FewRel dataset. The results are shown in the table below. All baseline results are directly quoted from [2]. CRL is the previous SOTA method for continual relation learning. $\dagger$ denotes [2]'s reproduced results with the open codebases. $*$ represents results based on our re-implementation using open-source code.
>
> | Model           | T1              | T2          | T3              | T4              | T5              | T6              | T7              | T8              | T9              | T10             |
> | --------------- | --------------- | --------------- | --------------- | --------------- | --------------- | --------------- | --------------- | --------------- | --------------- | --------------- |
> | EA-EMR          | $89.0$          | $69.0$          | $59.1$          | $54.2$          | $47.8$          | $46.1$          | $43.1$          | $40.7$          | $38.6$          | $35.2$          |
> | EMAR            | $88.5$          | $73.2$          | $66.6$          | $63.8$          | $55.8$          | $54.3$          | $52.9$          | $50.9$          | $48.8$          | $46.3$          |
> | CML             | $91.2$          | $74.8$          | $68.2$          | $58.2$          | $53.7$          | $50.4$          | $47.8$          | $44.4$          | $43.1$          | $39.7$          |
> | EMAR+BERT       | $\mathbf{98.8}$ | $89.1$          | $89.5$          | $85.7$          | $83.6$          | $84.8$          | $79.3$          | $80.0$          | $77.1$          | $73.8$          |
> | RP-CRE          | $97.9$          | $92.7$          | $91.6$          | $89.2$          | $88.4$          | $86.8$          | $85.1$          | $84.1$          | $82.2$          | $81.5$          |
> | RP-CRE$\dagger$ | $98.4$          | $95.2$          | $93.1$          | $91.4$          | $90.8$          | $88.8$          | $87.6$          | $86.8$          | $85.2$          | $83.9$          |
> | CRL             | $98.3$          | $\{95.4}$ | $\{93.4}$ | $\{92.0}$ | $\{91.0}$ | $\{89.7}$ | $\{88.3}$ | $\{87.0}$ | $\{85.6}$ | $\{84.4}$ |
> | CRL   + Our PFD*          | $98.6$          | $\mathbf{95.7}$ | $\mathbf{93.8}$ | $\mathbf{92.8}$ | $\mathbf{91.3}$ | $\mathbf{89.9}$ | $\mathbf{89.4}$ | $\mathbf{87.5}$ | $\mathbf{86.4}$ | $\mathbf{85.3}$ |
>
>
> Based on the results from the aforementioned table, our PFD consistently improves the performance of SOTA methods across various distinct NLP continual learning tasks. This is attributed to its skillful navigation of the balance between retaining old knowledge and acquiring new ones, thus more effectively addressing the challenge of catastrophic forgetting.
>
>
> Q2: The paper does not compare the proposed approach with other state-of-the-art methods outside the CNER domain, which may limit its generalizability to other NLP tasks.
>
> A2: Thank you for your insightful comment. We applied the state-of-the-art methods Snapshot[1], CRL[2], and Dynamic Knowledge Distillation[3] from domains continual text classification, continual relation learning, and continual machine translation, respectively, to the CNER domain. The experimental results on the I2B2 dataset under the setting FG-8-PG-2 are presented below. $*$ represents results based on our re-implementation using open-source code. Other baseline results are cited from CFNER[4].
>
> | Baseline | Mi-F1 | Ma-F1|
> | :------: | :------: | :------: |
> | FT | 23.60±0.15 | 23.54±0.38 |
> | PODNet |36.22±12.9 | 26.08±7.42
> | LUCIR |68.54±0.27 | 46.94±0.63|
> | ST |48.94±6.78 | 29.00±3.04|
> | Extend* | 56.48±2.41 | 38.88±1.38 |
> | Extend | 52.25±5.36  | 30.93±2.77 |
> | CFNER* |69.12±0.94 | 51.61±0.87 |
> | CFNER | 69.07±0.89 | 51.09±1.05 |
> | Snapshot* | 49.24±2.44 | 29.12±1.58 |
> | CRL* | 66.19 ± 0.92 | 49.83 ± 0.62|
> | Dynamic Knowledge Distillation* | 63.69±0.60 | 42.99±1.13 |
> |CPFD (Ours)| **81.05±0.87** | **65.04±1.13** |
>
>
> From the results in the above table, it is evident that the three methods, Snapshot, CRL, and Dynamic Knowledge Distillation, prioritize addressing the common issue of forgetting while overlooking the specific challenge of semantic shift in the CNER domain. As a consequence, their performance in the CNER field falls below that of our method.
>
>
> Q3: The paper does not provide a detailed discussion of the computational complexity of the proposed approach, which may be a concern for large-scale datasets or real-time applications.
>
> A3: Thank you for raising this important point. When confronted with the acquisition of a new task, the computational intricacies of our CPFD framework are contingent on two conditions: 1) In cases where the task count stands at 1 (i.e., $t=1$), the primary computational expense of CPFD lies in the updating of $\theta_t$ (where $t=1$). To be more precise, the expenditure associated with updating $\theta_t$ is denoted as $O(\xi(N,n_t))$, where the function $\xi(\cdot)$ hinges on the single-task learner (namely, the base model), and $n_t$ signifies the quantity of training samples within task $t$, while $N$ signifies the quantity of model parameters. Consequently, upon the advent of a new task, the comprehensive computational complexity inherent in our proposed model materializes as $O(\xi(N,n_t))$. In scenarios where the task count surpasses 1 (i.e., $t > 1$), the principal computational expense of assimilating a new task encompasses two distinct subproblems: the first optimization problem resides within Equation (3), while the second is encapsulated in Equation (6). For the problem in Eq (3), the cost of updating $\theta_t$ is $O(\xi(N, n_t))$. For the problem in Equation (6), the expenditure associated with calculating pseudo-labels $\widehat{Y}_{t-1}$ stands at $O(\zeta(N, n_t))$, with $\zeta(\cdot)$ denoting the inference procedure of the old model. This inference procedure encompasses a single neural network forward propagation, the computational overhead of which pales in comparison to the training process. The update of $\theta_t$ bears a cost of $O(\xi(N, n_t))$, whereas the computation associated with the adaptive weight $\eta$ incurs a cost of $O(n_t)$. The overall computational complexity of our proposed method is $O(\xi(N, n_t)+\zeta(N, n_t)+n_t)$.
>
>
> Q4: The authors need to tell the strengths and weaknesses of their approach and provide a balanced and nuanced discussion of their findings. And the paper does not provide a detailed analysis of the limitations of the proposed approach, such as cases where it may not work well or situations where it may be less effective.
>
> A4: Thank you for your thoughtful feedback.
>
> **Strengths:** **1) Catastrophic Forgetting Mitigation:** Our method incorporates a pooled features distillation loss, effectively mitigating catastrophic forgetting. This mechanism enables the retention of linguistic knowledge while striking a harmonious balance between stability and plasticity. **2) Semantic Shift Handling:** Our method introduces a confidence-based pseudo-labeling strategy to better recognize old entity types for the current non-entity type tokens and deal with the semantic shift problem. **3) Type Imbalance Handling:** To contend with imbalanced type distributions, our method proposes an adaptive re-weighting type-balanced learning strategy tailored for continual named entity recognition. **4) State-of-the-art performance:** The outcomes of our comprehensive experiments, conducted across ten different CNER settings on three distinct datasets, substantiate the effectiveness of our method. It surpasses the previous state-of-the-art methods, showcasing an average gain of 6.3\% in Micro F1 scores and 8.0\% in Macro F1 scores.
>
> **Weaknesses/Limitations:**  Due to the recognition errors of the old model, the pseudo-label strategy inevitably introduces some noisy labels. When the amount of noisy labels is substantial, it can potentially impact the model's performance. Although our confidence-based pseudo-labeling strategy can mitigate the recognition errors of the old model by employing a median entropy as a confidence threshold and retaining only those pseudo labels where the old model exhibits sufficient confidence, it is not completely immune to errors, and some mislabeled instances may still persist.
>
>
> **Reference**
>
> [1] Wang J, Dong D, Shou L, et al. Effective Continual Learning for Text Classification with Lightweight Snapshots[C]//Proceedings of the AAAI Conference on Artificial Intelligence. 2023, 37(8): 10122-10130.
>
> [2] Zhao K, Xu H, Yang J, et al. Consistent Representation Learning for Continual Relation Extraction[C]//Findings of the Association for Computational Linguistics: ACL 2022. 2022: 3402-3411.
>
> [3] Cao Y, Wei H R, Chen B, et al. Continual learning for neural machine translation[C]//Proceedings of the 2021 Conference of the North American Chapter of the Association for Computational Linguistics: Human Language Technologies. 2021: 3964-3974.
>
> [4] Zheng, Junhao, et al. "Distilling Causal Effect from Miscellaneous Other-Class for Continual Named Entity Recognition." Proceedings of the 2022 Conference on Empirical Methods in Natural Language Processing. 2022.

---

### Official Review · Reviewer_CtRi · 2023-08-05

**Soundness:** 4

**Excitement:**

3: Ambivalent: It has merits (e.g., it reports state-of-the-art results, the idea is nice), but there are key weaknesses (e.g., it describes incremental work), and it can significantly benefit from another round of revision. However, I won't object to accepting it if my co-reviewers champion it.

**Missing References:**

[1] Pelosin F, Jha S, Torsello A, et al. Towards exemplar-free continual learning in vision transformers: an account of attention, functional and weight regularization[C]//Proceedings of the IEEE/CVF Conference on Computer Vision and Pattern Recognition. 2022: 3820-3829.

**Paper Topic And Main Contributions:**

To alleviate the stability-plasticity dilemma common in continual learning, pooled features distillation is adopted.
To address the semantic shift problem specific to CNER, this paper filters low-confidence pseudo labels and assigns different weights to tokens of different types based on the number of tokens.


**Questions For The Authors:**

Question A: Quantification analysis of the stability-plasticity trade off, similar to the Stability-Plasticity Curves in [1], is expected to verifies the claim that “our method establishes a suitable balance between stability and plasticity”.

**Reasons To Accept:**

The paper is well-written and experimental results verifies the effectiveness.

**Reasons To Reject:**

This paper adopts similar attention regularization and uncertainty-aware distillation such as [1][2] to alleviate the stability-plasticity dilemma and semantic shift proposed by [3].

[1] Pelosin F, Jha S, Torsello A, et al. Towards exemplar-free continual learning in vision transformers: an account of attention, functional and weight regularization[C]//Proceedings of the IEEE/CVF Conference on Computer Vision and Pattern Recognition. 2022: 3820-3829.
[2] Kurmi, Vinod K., et al. "Do not forget to attend to uncertainty while mitigating catastrophic forgetting." Proceedings of the IEEE/CVF Winter Conference on Applications of Computer Vision. 2021.
[3] Zheng, Junhao, et al. "Distilling Causal Effect from Miscellaneous Other-Class for Continual Named Entity Recognition." Proceedings of the 2022 Conference on Empirical Methods in Natural Language Processing. 2022.


**Reproducibility:**

4: Could mostly reproduce the results, but there may be some variation because of sample variance or minor variations in their interpretation of the protocol or method.

**Reviewer Confidence:**

4: Quite sure. I tried to check the important points carefully. It's unlikely, though conceivable, that I missed something that should affect my ratings.

---

> ### Author Rebuttal · Authors · 2023-08-27
>
> Thank you for your constructive comments. We would like to address your concerns and answer your questions in the following.
>
> Q1: This paper adopts similar attention regularization and uncertainty-aware distillation such as [1][2] to alleviate the stability-plasticity dilemma and semantic shift proposed by [3].
>
> A1: Thank you for your inquiry. Our distillation approach and the attention regularization presented in [1] both involve enhancing the degrees of freedom of the distillation loss by first performing pooling operations on the attention map with dimensions (head, height, width). This is done to alleviate the stability-plasticity dilemma. However, [1] only performs pooling on the attention map along the height and width dimensions. In addition, we conduct pooling along the head dimension. The multiple heads in self-attention can be likened to partitioning a large feature space into several distinct and non-overlapping sub-feature spaces. Subsequently, attention calculations are performed in parallel within each of these subspaces. This aids the network in capturing a more diverse set of representational information, bearing some resemblance to the concept of channels in CNNs. Pooling along the head dimension helps in disregarding unimportant noise information and distills crucial information from different subspaces. Here, we conduct supplementary ablation experiments on the I2B2 and OntoNotes5 datasets under the setting FG-1-PG-1, as shown in the table below. The results indicate that compared to performing attention map pooling along only the height and width dimensions followed by distillation ($L_{PFD\ w/\ height+width}$ [1]), performing attention map pooling along the head, height, and width dimensions followed by distillation ($L_{PFD\ w/\ head+height+width}$ (Ours)) can lead to improved CNER performance.
>
> | Distillation Loss| I2B2 dataset|  |OntoNotes5 dataset |  |
> | :-----: | :----: | :----: |:----: | :----: |
> | | Mi-F1 | Ma-F1 |  Mi-F1 | Ma-F1 |
> |$L_{PFD\ w/\ head+height+width}$ (Ours) | **74.19±0.95**  | **48.34±1.45**  | **66.73±0.70**  | **54.12±0.30** |
> |$L_{PFD\ w/\ height+width}$ [1] |72.95±0.89| 47.01±1.11| 65.07±1.49| 52.77±0.85|
>
> [2] proposes uncertainty-aware distillation to help the model preserve useful information from the previous tasks, not specifically to alleviate the semantic shift issue. The issue refers to the scenario in the CNER paradigm where tokens labeled as the non-entity type could encompass: the actual non-entity type, previously learned old entity types, or future ones that have not been encountered yet. To address this issue, our confidence-based pseudo-labeling strategy explicitly retrieves old entity types within the current non-entity type via the old model for classification. To better reduce the recognition errors from the old model, it employs entropy as a measure of uncertainty and the median entropy as a confidence threshold, retaining only those pseudo labels where the old model exhibits sufficient confidence.
>
>
> Although [3] was the first to identify the semantic shift problem in CNER, it didn't effectively address this concern. There are two significant differences between their approach and ours. Firstly, [3] employs the old model to identify non-entity type tokens belonging to old entity types for distilling causal effect. In contrast, we explicitly retrieve old entity types within the current non-entity type for classification. Our experimental results in Table 1 of the paper demonstrate that the latter approach is more effective. Secondly, [3] does't sufficiently handle recognition errors by the old model. While they introduce a curriculum learning strategy to mitigate recognition errors, this strategy requires the manual pre-definition of numerous hyper-parameters, limiting its applicability and the effect of error reduction. In contrast, our confidence-based pseudo-labeling strategy employs median entropy as the confidence threshold, eliminating the need for manually predefining any hyper-parameters, better reducing the recognition error from the old model, and dealing with the semantic shift problem.
>
>
>
>
> Q2:  Quantification analysis of the stability-plasticity trade off, similar to the Stability-Plasticity Curves in [1], is expected to verifies the claim that “our method establishes a suitable balance between stability and plasticity”.
>
> A2: Thank you for your valuable suggestion. Following the definition in [1], we have plotted the task-aware Stability-Plasticity Curves for the FG-1-PG-1 settings of the I2B2 and Ontonotes5 datasets to compare the stability-plasticity dilemma for various methods, including Ours w/ $L_{FD}$, Ours w/ $L_{PFD-lax}$, and Ours (equipped with $L_{PFD}$ by default). The specific definitions of these three distillation losses can be found in Section 4.1 of our paper. $L_{FD}$ is a rigid loss, which compels the new model to strictly align with the attention weights found in the old model. This practice tends to infuse the model with excessive stability (i.e., preservation of old knowledge), thereby compromising its plasticity (i.e., ability to absorb new knowledge). This is reflected in the curves, where as the number of incremental tasks increases, the stability score remains consistently high, while the plasticity score for each task is relatively low. $L_{PFD-lax}$ is a permissive loss, which utilizes a more aggressive pooling to lessen the similarity between the new model and the old model. In contrast to Ours w/ $L_{FD}$, Ours w/ $L_{PFD-lax}$'s curve looks like this: as the number of incremental tasks increases, its stability score gradually decreases, while the plasticity score for each task is relatively high. Finally, $L_{PFD}$ achieves an appropriate balance between excessive stability and extreme plasticity via adopting a less aggressive pooling. From the curves, it is evident that as the number of incremental tasks increases, Ours consistently achieves the highest scores for both stability and plasticity. Due to the inconvenience of inserting images in the response box, we will add these task-aware Stability-Plasticity curve graphs in the final version.
>
>
> Q3: Missing References [1].
>
> A3: Thank you for bringing this to our attention. We apologize for the oversight in not including the reference [1] you mentioned. We will include this reference [1] in the appropriate section of the final version. Your diligence in reviewing our work is greatly appreciated.
>
>
> **Reference**
>
> [1] Pelosin F, Jha S, Torsello A, et al. Towards exemplar-free continual learning in vision transformers: an account of attention, functional and weight regularization[C]//Proceedings of the IEEE/CVF Conference on Computer Vision and Pattern Recognition. 2022: 3820-3829.
>
> [2] Kurmi, Vinod K., et al. "Do not forget to attend to uncertainty while mitigating catastrophic forgetting." Proceedings of the IEEE/CVF Winter Conference on Applications of Computer Vision. 2021.
>
> [3] Zheng, Junhao, et al. "Distilling Causal Effect from Miscellaneous Other-Class for Continual Named Entity Recognition." Proceedings of the 2022 Conference on Empirical Methods in Natural Language Processing. 2022.

---

### Meta-Review · Area_Chair_ybad · 2023-09-19

**Recommendation:** 4

**Metareview:**

Summary:
This paper addresses two challenges in Chinese Named Entity Recognition (CNER): catastrophic forgetting and non-entity type semantic transfer issues. To tackle these problems, it introduces a phased approach with corresponding strategies. This includes the introduction of an aggregated feature distillation loss to balance stability and plasticity, effectively mitigating catastrophic forgetting. Additionally, it proposes a confidence-based pseudo-labeling plan to explicitly extract old entity types within non-entity types, reducing label noise's impact and handling semantic transfer. The significance of component collaboration in solving CNER is validated through ablation experiments.

Reason To Accept:
1.The motivation behind this work is clear: it addresses the semantic shift problem in Continual Named Entity Recognition (CNER), a challenge overlooked by previous methods.
2.The paper's strengths lie in its innovative approach to tackling catastrophic forgetting in CNER.
3.The paper is well-written, and its experimental results validate its effectiveness. It offers a comprehensive comparison of the proposed method with other baselines across various settings and datasets. Extensive experiments conducted on ten CNER settings involving three datasets showcase the approach's effectiveness and significant improvements over previous state-of-the-art methods. The proposed approach and experimental results have the potential to inspire further research and enhance performance in CNER and related tasks.

Reason To Reject:
1.This paper adopts similar attention regularization and uncertainty-aware distillation such as [1][2] to alleviate the stability-plasticity dilemma and semantic shift proposed by [3]. ([1] Pelosin F, Jha S, Torsello A, et al. Towards exemplar-free continual learning in vision transformers: an account of attention, functional and weight regularization[C]//Proceedings of the IEEE/CVF Conference on Computer Vision and Pattern Recognition. 2022: 3820-3829. [2] Kurmi, Vinod K., et al. "Do not forget to attend to uncertainty while mitigating catastrophic forgetting." Proceedings of the IEEE/CVF Winter Conference on Applications of Computer Vision. 2021. [3] Zheng, Junhao, et al. "Distilling Causal Effect from Miscellaneous Other-Class for Continual Named Entity Recognition." Proceedings of the 2022 Conference on Empirical Methods in Natural Language Processing. 2022.)
2.The experiments are inadequate. 1) The paper only conducts experiments on BERT, leaving the performance on larger models unexplored. 2) The proposed approach may not be universally applicable to all NLP tasks and datasets, with its effectiveness potentially contingent on specific data characteristics. 3) The paper lacks comparisons of the proposed approach with other state-of-the-art methods outside the CNER domain, limiting its generalizability to other NLP tasks.
3.Certain details and analyses are lacking or unclear. 1) The performance under different hyper-parameter settings (e.g., \lambda) is unclear. 2) A comprehensive analysis of the proposed approach's limitations, including scenarios where it might perform poorly or be less effective, is missing. 3) There's no detailed discussion regarding the computational complexity of the proposed approach, which could be a concern for large-scale datasets or real-time applications. The authors should provide a balanced and nuanced discussion of both the strengths and weaknesses of their approach.

---

### Decision · Program_Chairs · 2023-10-07

**Decision:**

Accept-Main

**Comment:**

Summary:
This paper addresses two challenges in Chinese Named Entity Recognition (CNER): catastrophic forgetting and non-entity type semantic transfer issues. To tackle these problems, it introduces a phased approach with corresponding strategies. This includes the introduction of an aggregated feature distillation loss to balance stability and plasticity, effectively mitigating catastrophic forgetting. Additionally, it proposes a confidence-based pseudo-labeling plan to explicitly extract old entity types within non-entity types, reducing label noise's impact and handling semantic transfer. The significance of component collaboration in solving CNER is validated through ablation experiments.

Reason To Accept:
1.The motivation behind this work is clear: it addresses the semantic shift problem in Continual Named Entity Recognition (CNER), a challenge overlooked by previous methods.
2.The paper's strengths lie in its innovative approach to tackling catastrophic forgetting in CNER.
3.The paper is well-written, and its experimental results validate its effectiveness. It offers a comprehensive comparison of the proposed method with other baselines across various settings and datasets. Extensive experiments conducted on ten CNER settings involving three datasets showcase the approach's effectiveness and significant improvements over previous state-of-the-art methods. The proposed approach and experimental results have the potential to inspire further research and enhance performance in CNER and related tasks.

Reason To Reject:
1.This paper adopts similar attention regularization and uncertainty-aware distillation such as [1][2] to alleviate the stability-plasticity dilemma and semantic shift proposed by [3]. ([1] Pelosin F, Jha S, Torsello A, et al. Towards exemplar-free continual learning in vision transformers: an account of attention, functional and weight regularization[C]//Proceedings of the IEEE/CVF Conference on Computer Vision and Pattern Recognition. 2022: 3820-3829. [2] Kurmi, Vinod K., et al. "Do not forget to attend to uncertainty while mitigating catastrophic forgetting." Proceedings of the IEEE/CVF Winter Conference on Applications of Computer Vision. 2021. [3] Zheng, Junhao, et al. "Distilling Causal Effect from Miscellaneous Other-Class for Continual Named Entity Recognition." Proceedings of the 2022 Conference on Empirical Methods in Natural Language Processing. 2022.)
2.The experiments are inadequate. 1) The paper only conducts experiments on BERT, leaving the performance on larger models unexplored. 2) The proposed approach may not be universally applicable to all NLP tasks and datasets, with its effectiveness potentially contingent on specific data characteristics. 3) The paper lacks comparisons of the proposed approach with other state-of-the-art methods outside the CNER domain, limiting its generalizability to other NLP tasks.
3.Certain details and analyses are lacking or unclear. 1) The performance under different hyper-parameter settings (e.g., \lambda) is unclear. 2) A comprehensive analysis of the proposed approach's limitations, including scenarios where it might perform poorly or be less effective, is missing. 3) There's no detailed discussion regarding the computational complexity of the proposed approach, which could be a concern for large-scale datasets or real-time applications. The authors should provide a balanced and nuanced discussion of both the strengths and weaknesses of their approach.